# Caspase-8 and Tyrosine Kinases: A Dangerous Liaison in Cancer

**DOI:** 10.3390/cancers15133271

**Published:** 2023-06-21

**Authors:** Claudia Contadini, Alessandra Ferri, Claudia Cirotti, Dwayne Stupack, Daniela Barilà

**Affiliations:** 1Laboratory of Cell Signaling, IRCCS-Fondazione Santa Lucia, 00179 Rome, Italy; claudiacirotti89@gmail.com; 2Department of Biology, University of Rome “Tor Vergata”, 00133 Rome, Italy; 3Department of Pathology and Laboratory Medicine, New York Presbyterian Hospital, Weill Cornell Medicine, New York, NY 10021, USA; alf4012@med.cornell.edu; 4Moores Cancer Center, University of California San Diego, 3855 Health Sciences Drive, La Jolla, CA 92093-0803, USA; dstupack@health.ucsd.edu

**Keywords:** Caspase-8, tyrosine kinase, cancer, cell cycle, NF-kB signaling

## Abstract

**Simple Summary:**

Caspase-8 is a protease mediating the activation of the extrinsic apoptotic process that leads to programmed cellular death. Evasion of apoptosis is one of the key hallmarks of cancer and Caspase-8 has commonly been associated with an antitumor protective role. However, observations that several solid tumor types inconsistently display aberrantly high levels of Caspase-8 has fueled studies that challenge this dogma. In this review, we summarize the current state of the art on how tumors benefit from high levels of Caspase-8 expression. In addition, we discuss the mechanisms through which tumors are able to alter the function of Caspase-8 and turn a protective protein into an ally. Specifically, we focus on the role played by tyrosine kinases in inhibiting the enzymatic role of Caspase-8 and remodulating Caspase-8 function in cancer through tyrosine phosphorylation.

**Abstract:**

Caspase-8 is a cysteine-aspartic acid protease that has been identified as an initiator caspase that plays an essential role in the extrinsic apoptotic pathway. Evasion of apoptosis is a hallmark of cancer and Caspase-8 expression is silenced in some tumors, consistent with its central role in apoptosis. However, in the past years, several studies reported an increased expression of Caspase-8 levels in many tumors and consistently identified novel “non-canonical” non-apoptotic functions of Caspase-8 that overall promote cancer progression and sustain therapy resistance. These reports point to the ability of cancer cells to rewire Caspase-8 function in cancer and raise the question of which are the signaling pathways aberrantly activated in cancer that may contribute to the hijack of Caspase-8 activity. In this regard, tyrosine kinases are among the first oncogenes ever identified and genomic, transcriptomic and proteomic studies indeed show that they represent a class of signaling molecules constitutively activated in most of the tumors. Here, we aim to review and discuss the role of Caspase-8 in cancer and its interplay with Src and other tyrosine kinases.

## 1. Introduction

Caspase-8 (FLICE) was originally identified as a cysteine protease recruited to the CD95 (Fas/APO-1) death inducing signaling complex (DISC) [1]. Several studies confirmed the crucial role of Caspase-8 in apoptosis triggered by Fas and by other death receptors, including TRAIL receptors (DR4 and DR5), and clarified the molecular mechanisms that allow its activation in these signaling pathways. Upon death receptor stimulation by their relative ligands, Caspase-8 is recruited and participated in the assembly of the Death-Inducing Signaling Complex (DISC) [1,2]. This event is essential to drive the enzymatic activation of Caspase-8. Indeed, Caspase-8 is considered an initiator caspase that is produced as a proenzyme: the N-terminal region presents two DED domains, followed by large (p20/p18) and small (p12/p10) subunits. The conversion from the proenzyme to the fully active enzyme is promoted by its recruitment to the DISC, which allows Caspase-8 dimerization, priming a series of autoprocessing events at specific aspartic acid residues that culminate with the release of the large and small subunits [3]. This event is essential to achieve the assembly of the fully active tetrameric Caspase-8 complex, formed by two large and two small subunits, and to ensure its release from the DISC and its ability to cleave its substrates, which are key events to initiate and execute the canonical extrinsic apoptotic cascade [2]. 

The central role of Caspase-8 in the apoptotic response triggered by death receptor stimulation and the clear association between the enzymatic activity of several caspases and the induction and execution of cell death have strongly enforced the idea that the main function of Caspase-8 is linked to its canonical role in apoptosis. Nonetheless, studies by different laboratories identified several non-apoptotic functions of Caspase-8 [4,5,6]. Caspase-8 and the death receptors seem to have evolved along independent timelines, suggesting that non-apoptotic roles could represent key ancient conserved functions or simply vestigial activity [7]. In this regard, we will focus on the role of Caspase-8 in cancer and we will discuss studies aimed at uncovering its multiple functions in cancer development and in the response to therapy. The molecular mechanisms that may modulate the function and the activity of Caspase-8 in cancer are still largely obscure; however, the interplay between Caspase-8 and the aberrant tyrosine phosphorylation signaling that characterizes most tumors is emerging as a major character. 

## 2. Caspase-8 in Cancer

Evasion of apoptosis is a well-established hallmark of cancer and contributes both to cancer initiation and development, as well as to cancer chemo- and radiotherapy resistance. In this regard, Caspase-8 expression has been reported to be downregulated through promoter methylation in some tumors, including neuroblastoma, and mutated in some others [6,8].

Surprisingly, Caspase-8 expression is retained and even increased in some tumors compared to normal tissue (reviewed in [8,9]). This raises two main questions: (1) Do cancer cells rewire Caspase-8 function? (2) Which molecular mechanisms switch Caspase-8 function in cancer? 

### Modulation of Caspase-8 Expression Levels and Caspase-8 Mutations in Cancer

The expression levels of Caspase-8 have been largely investigated in many tumors. The availability of genomic and transcriptomic data allowed reporting of significant variations in Caspase-8 expression in different cancer and these changes have been recently reviewed [8]. Here, based on an analysis of the literature, we distinguish tumors in which Caspase-8 expression has been reported to be aberrantly downregulated from those that conversely upregulate Caspase-8 expression compared to normal tissue. Caspase-8 expression is decreased in neuroblastoma, small cell lung carcinoma, brain tumors such as medulloblastoma and glioblastoma, liver, breast, prostate, stomach and ovary tumors. Conversely, Caspase-8 expression is increased in colorectal, cervical and renal cancers compared to normal tissues [8]. This report is partially divergent from others. When comparing the relative expression of Caspase-8 protein in normal and cancer tissues, Caspase-8 levels have been reported to be aberrantly low in kidney, prostate, colorectal and breast tumors, and conversely, unexpectedly high in glioma, cervix, pancreas and liver cancer [9]. In addition, Muller et al. performed a bioinformatic analysis of TCGA data and, according to their analysis, Caspase-8 expression is higher in renal carcinoma, gastric adenocarcinoma, hepatocellular carcinoma, glioblastoma, lung adenocarcinoma, urothelial carcinoma, prostate adenocarcinoma and melanoma [10]. Of note, regarding glioblastoma, high levels of Caspase-8 expression have been proposed to be part of the molecular signature that identified the mesenchymal subtype [11]. More interestingly, the correlation between the levels of Caspase-8 expression and glioblastoma patients’ survival has also been investigated. In this regard, we reported that higher levels of Caspase-8 correlate with a worse prognosis [12].

Furthermore, we queried twenty cancer databases from The Cancer Genome Atlas (TCGA) with overall survival data to determine whether the impact of Caspase-8 expression was a significant predictor of improved or worsened overall survival by Kaplan–Meier analysis (Figure 1). Interestingly, from this analysis, we observed that Caspase-8 can be either pro- or antitumor, but rarely “neutral”, highlighting again the importance of this protein in cancer progression. 

Regarding the molecular mechanisms that drive the downregulation of Caspase-8 expression in cancer and eventually also affect its functionality, several reports identified the occurrence of inactivating mutations of Caspase-8 in tumors (reviewed in [8]). Inactivating mutations of Caspase-8 have been reported in colorectal carcinoma [13] and in head and neck squamous cell carcinoma [14,15,16].

An alternative mechanism to drive Caspase-8 downregulation in cancer relies on Caspase-8 promoter epigenetic silencing. The hypermethylation of Caspase-8 promoter and the subsequent loss of Caspase-8 expression have been reported in medulloblastoma [17,18], neuroblastoma [19], cervical cancer [20], breast cancer [21] and glioblastoma [22].

Certainly, a loss of Caspase-8 expression or function in cancer cells may support evasion from apoptosis and therefore promote cancer progression and resistance to radio and chemotherapy.

Conversely, Caspase-8 upregulation in cancer is an event that is usually linked to the aberrant activation of molecular switches that prevent the full induction of Caspase-8 proteolytic activity leading to apoptosis, and the concomitant firing of novel alternative functions of Caspase-8 that support the cancer phenotype. Examples of molecular switches include the upregulation of FLIP proteins, which attenuate the activation of Caspase-8 [23,24], and the activation of kinase signaling, which promotes Caspase-8 phosphorylation [25,26]. The interplay between FLIP and Caspase-8 in cancer has been largely reviewed elsewhere [24,27]. Here, we will focus mainly on the role of Caspase-8 phosphorylation and on the acquired ability of cancer cells to redirect Caspase-8 to sustain tumor progression and resistance to therapy.

## 3. Non-Apoptotic Functions of Caspase-8 in Cancer

### 3.1. Caspase-8 Modulates Cell Adhesion and Migration

Beside the canonical role in programmed cell death, Caspase-8 also plays a role in cytoskeletal remodeling, cell adhesion, and cell migration. It acts as a part of distinct environmental biosensor complexes in the periphery of the cells, where it sustains migration or cell death depending on the stimuli received [28]. This non-apoptotic function has been shown to rely on the ability of Caspase-8 to act as a scaffold or adaptor for the formation of specific protein complexes, rather than on its enzymatic activity. Specifically, Caspase-8 interacts with several components of the focal adhesion complex in a tyrosine kinase-dependent manner and promotes calpain protease activity and calpain-dependent processes (i.e., Rac activation, lamellipodia assembly). In this way, Caspase-8 promotes cytoskeletal remodeling [29] and focal adhesion turnover and integrin recycling, thereby sustaining both cell migration in vitro and metastasis in vivo among apoptosis-resistant tumors [30,31]. The interaction of Caspase-8 with calpain occurs in both human and murine cells, suggesting that the event does not require phosphorylation on Tyrosine 380 (Tyr380), the most common target site for Tyr phosphorylation, which is not conserved in mouse Caspase-8.

### 3.2. Caspase-8 Modulates NF-kB Signaling and Inflammation in Cancer Progression and Therapy 

The role of Caspase-8 in the inflammatory process is important in cells of the immune system, where its interplay with the transcription factor NF-κB is extensively studied in lymphocyte activation and in macrophage differentiation [32,33,34]. Indeed, Caspase-8 can regulate NF-κB activation both in a proteolysis-dependent and proteolysis-independent manner, depending on the cell type.

The first evidence of a correlation between Caspase-8 and NF-κB was observed in overexpression studies in the HEK293 cell line, where Caspase-8 protein was shown to promote NF-κB activation in a manner independent of its catalytic activity [35]. In addition, in human bone marrow-derived mesenchymal stromal cells, both Caspase-8 silencing and z-IETD (a selective Caspase-8 inhibitor) impair NF-κB nuclear accumulation and pro-inflammatory cytokines released upon LPS stimulation [36], suggesting that in this context Caspase-8 enzymatic activity is important to modulate NF-κB activity in response to TLR. Several works supported a scaffolding role of Caspase-8 in the promotion of NF-κB signaling and in cytokines release in response to inflammatory stimuli, such as TCR ligands or TRAIL [37,38,39].

In cancer, the role of Caspase-8 in the promotion of inflammation is still controversial. This is understandable, since the impact of Caspase-8 expression appears to be remarkably different based upon tumor context. Depending on cancer type, the expression of Caspase-8 may be up- or downregulated. The heterogeneity in the genetic and epigenetic alterations of Caspase-8 in cancer, as well as in its function, represent crucial considerations in predicting the therapy response.

Although an overt loss of Caspase-8 is commonly observed in several tumors, which is consistent with its canonical apoptotic function, increased Caspase-8 expression has been reported in many other tumors [9]. High levels of Caspase-8 are associated with poor prognosis in patients with glioma [12], hepatocellular carcinoma [40] and pancreatic cancer [41]. Low Caspase-8 levels are associated with a worse prognosis in patients with neuroblastoma [31,42,43], neuroendocrine lung tumors [44] and gynecological tumors, in which Caspase-8 loss promotes tumor aggressiveness and invasiveness [45].

For example, in ovarian cancer, low Caspase-8 expression levels correlate with chronic inflammation, immunoediting, and immune resistance, thereby sustaining tumor aggressiveness. Indeed, Caspase-8 plays an antitumorigenic role in the primary tumor cells and in the tumor microenvironment (TME) by regulating B and T lymphocyte activation and macrophage differentiation and polarization [5].

Among the tumors in which Caspase-8 expression are upregulated, glioblastoma (GBM) shows ex novo synthesis of Caspase-8 protein. In this context, Caspase-8 plays a pro-tumorigenic role, promoting sustained NF-κB activation, inflammation and angiogenesis [12,46]. Accordingly, the level of Caspase-8 expression correlates with high levels of inflammatory factors, such as IL-8, IL-6, IL1β, CCL2/MCP1 and VEGF, in the TME [12]. Interestingly, we observed that the phosphorylation of Caspase-8 on Tyrosine 380, a well-known residue phosphorylated by Src kinase, sustains the Caspase-8–NF-κB axis in GBM [46]. This Src-mediated phosphorylation is known to inhibit Caspase-8 activity [25] and to sustain the interaction with Src homology domain 2 (SH2) proteins and with the p85 subunit of phosphoinositide 3-kinase, thereby promoting cell migration and invasion [47,48]. Interestingly, we recently observed that Caspase-8 phosphorylation on Tyr380 is necessary for the interaction of Caspase-8 with IKK proteins and NF-κB, promoting NF-κB nuclear localization and the release of inflammatory and angiogenetic factors in glioblastoma cells [46].

### 3.3. Caspase-8 Influence on Cell Cycle Control

The newly emerging non-canonical roles of Caspase-8 have challenged the dogma that identifies the cysteine-aspartic protease as a classic tumor suppressor due to its role in programmed cell death. As discussed above, in vitro and in vivo evidence have recognized Caspase-8 in cancer as both an enhancer of cell motility and migration [28,31,49], a promoter of tumorigenesis [12] and a sustainer of the increased inflammatory tumor microenvironment [46]. Additional non-apoptotic roles of Caspase-8 are emerging continuously. Here, we focus our attention on the potential impact of Caspase-8 on cell cycle control. 

Cancer cells are characterized by an extraordinary proliferative capacity and insensitivity to growth arrest signals [50]. The cellular machinery regulating cell proliferation is composed primarily by cyclins, cyclin-dependent kinases (CDK) and tumor suppressor genes as the retinoblastoma protein (pRb) and p53 [50]. The retinoblastoma protein is responsible for preventing excessive cell division by regulating the G1/S cell cycle checkpoint at which cells decide whether to progress into mitosis or go into the quiescent state. When the cell is not actively dividing, the pRb is bound to the transcription factor E2F; however, when appropriate signals for cell division are present, pRb is phosphorylated by cyclin E/CDK2 complex and releases E2F, which upregulates genes necessary for DNA replication and cell cycle progression [51]. TP53 controls appropriate cell division similarly to pRb; however, its functions are not limited to the G1/S checkpoint but instead encompass other checkpoints such as G2/M. Among cells harboring DNA damage, p53 acts to arrests cell division via p21 induction, which inhibits all cyclin-CDK complex formation and either signals for DNA repair or triggers apoptosis [52]. The loss of pRb or p53 function leads to the accumulation of DNA mutations and favors cancer onset. There is evidence that indicates Caspase-8 may aid cancer escape cell cycle control by influencing cell cycle machinery components such as pRb and p53 and disrupting cell cycle regulation [53]. 

Broadly, members of the Caspase family have been studied and identified as key regulators, both positive and negative, of the cell cycle [53]. Caspases can influence the cell cycle by cleaving cell cycle regulators such as p21, p27 or pRb [54,55], but are also known substrates of cell cycle kinases, e.g., CDK1 [53,56]. The first evidence of a non-apoptotic role of Caspase-8 in cell cycle control was identified by multiple studies in T-cells that showed Caspase-8 knock-out results in reduced S-phase entry and cellular proliferation [57,58]. Caspase-8 loss correlated with lower phosphorylation of the ribosomal protein S6 upon CD3 stimulation and reduced CDK2 activity, which ultimately compromise the capacity of cells to enter the S-phase of the cell cycle [57]. Beyond T-cells, Caspase-8 deficiency has been also linked to impaired S-phase entry in hepatocytes stimulated with epidermal growth factor (EGF) [59] and in which Caspase-8 knockdown in MDA-MB-231 breast cancer cells dramatically reduces cell proliferation, despite having little influence on cell viability [8,60].

Importantly, given the increase in cancer-related functions specific for Caspase-8, we hypothesize Caspase-8 may influence cancer progression by also impacting on cell cycle checkpoints. Frequently, cancer cells suffering genotoxic stress are arrested after the S-phase at the G2/M checkpoint [61], where p53 acts to either stimulate DNA damage repair or induce p53-dependent apoptosis [52]. More than 50% of all tumors harbor a mutation in p53 enabling them to overcome p53-dependent checkpoint regulation. Broadly, p53 mutations are commonly missense mutations leading to the incapacity of p53 to activate its canonical target genes, altering cell’s transcriptome favorably for cancer cells. Mutated p53 forms are selected for in response to tumor-induced stressed conditions providing cancer cells with a strong tool to overcome some of the main obstacles encountered by the tumor, such as: high levels of DNA damage caused by hyperproliferation of cancer cells, the presence of a strongly oxidative micro-environment, evading the antitumor response [52]. Interestingly, tumors harboring wild-type p53 are still able to progress and escape p53 control through alternative mechanisms that suppress wild-type p53 function [62,63]. The contrast between the incidence of cells with mutations in p53 and those with mutations in Caspase-8 (<1%) is striking, prompting questions about the overall role of Caspase-8 as a “tumor suppressor” [64,65].

A recent mechanism employing Caspase-8 to overcome the G2/M checkpoint has been identified by Müller et al. [10]. Using melanoma as a cancer model, the authors demonstrate Caspase-8 can influence cell cycle progression upon DNA damage. Genotoxic agents as UVB, temozolomide (TMZ) or cisplatin induce Caspase-8 translocation to the nucleus where it influences cell cycle dynamics by altering p53 levels. Specifically, their experiments demonstrate that nuclear Caspase-8 can cleave and induce the degradation of the ubiquitin carboxyl-terminal hydrolase 28 (USP28). Physiologically, USP28 is responsible for regulating p53 levels by removing ubiquitin molecules from p53 and preventing its degradation. However, in cancers harboring aberrant levels of Caspase-8, increased nuclear Caspase-8 localization results in USP28 cleavage and inactivation that consequently results in p53 proteosomal degradation [66]. This promotes cell proliferation and resistance to DNA damaging therapies.

Considering the effect of elevated Caspase-8 on p53 levels, the authors investigated whether Caspase-8 could also influence p53-controlled cell cycle proteins. Consistent with previous observations, they showed that Caspase-8 increases the capacity of cells to enter mitosis [10]. The activation of the mitosis-promoting factor CDK1 and of its co-factor cyclin B1 is reduced in Caspase-8 knockdown cells as compared to mitotic cells (MSO). In addition, the authors observed an upregulation in the levels of Polo-like kinase 1 (PLK1), a marker of mitotic prophase, in cells overexpressing Caspase-8 [10]. Using HeLa cells as a model, the authors confirm that cancer cells lacking Caspase-8 are deficient in proper cell division. Importantly, both Caspase-8 depletion and treatment with the pharmacologic inhibitor z-IETD reduced the number of cells undergoing mitotic spindle formation, once again stressing the importance of Caspase-8 enzymatic activity for its role in cell cycle control [10]. 

However, whether the activity of Caspase-8 is needed for its influence on cell cycle regulation is still unclear, as there is contrasting evidence in the literature supporting the need for its enzymatic activity or pointing towards an exclusively structural role of Caspase-8.

One of the first indications that Caspase-8 plays a role in regulating the cell cycle suggesting a scaffolding role for Caspase-8 in cancer, came from Boege et al., who uncovered Caspase-8 as a key component of DNA damage sensing in cancer [67]. Specifically, Boege et al. revealed that, in hepatocellular carcinoma (HCC), Caspase-8 has a non-apoptotic scaffolding role that is essential for DNA damage sensing and subsequent H2AX phosphorylation. The influence of Caspase-8 on DNA repair pathways points to a role in cell cycle regulation. According to Boege et al., and similarly to other non-canonical roles of Caspase-8 [46,49], its ability to influence DNA sensing is independent of its catalytic activity. Indeed, the authors demonstrated that Caspase-8 influences the capacity of cells in DNA damage recognition by forming a complex with RIPK1/FADD/cFLIP [67]. They show in vivo that mice harboring a Caspase-8 deletion in hepatocytes fail to activate the DNA damage response (DDR) pathway and to phosphorylate the histone H2AX, even upon detection of DNA damage [67]. 

In contrast, the importance of Caspase-8 enzymatic activity was once again stressed by Liccardi et al., who underline how cleaving activity is essential for its role in DNA damage sensing and cell cycle regulation [68]. The authors show the unusual formation of the ripoptosome complex (RIPK1/FADD/Caspase-8/cFLIP) during mitosis, supporting an additional role for the ripoptosome in the cell cycle other than in regulating the balance between apoptosis, inflammation and necroptosis. Treatment with the Caspase-8 inhibitors QVD and z-VAD induced an increase in the number of chromosomal alignment defects at the metaphase plate and accentuated abnormalities at the anaphase [68]. In accordance with their results, previous literature reports that the formation of the ripoptosome and a sub-lethal activation of Caspase-8 is not sufficient to induce of cellular death [69,70]. According to Liccardi et al., RIPK1 and Caspase-8 cooperate to respectively recruit and cleave PLK1, the kinase involved in regulating spindle assembly checkpoint and in supervising chromosomal segregation [68,71,72]. 

Altogether, this evidence stresses an under-researched connection between Caspase-8 and the ability of cancer cells to sense DNA damage and arrest the cell cycle, which opens the possibility to further understand the mechanisms that help cancer combat genotoxic stress and progress undisturbed. Further studies are needed to better clarify the mechanisms through which Caspase-8 impacts cell cycle control and whether or not its enzymatic activity is required for its function as a cell cycle regulator.

## 4. Molecular Mechanisms That Allow Cancer Cells to Rewire Caspase-8 Function

### 4.1. Role of Phosphorylation on Caspase-8

Given the role of Caspase-8 in the apoptotic pathway, its catalytic activity has to be finely regulated to avoid massive activation and ensure tissue homeostasis. Different mechanisms, including the expression of FLICE-like inhibitory protein (FLIP) family proteins and post-translational modifications (PTMs) such as ubiquitination and phosphorylation, occur concurrently to tightly regulate Caspase-8 activity [6]. Among PTMs, phosphorylation has a critical role in regulating protein activity and in pathological contexts, such as cancer, where the aberrant activation of tyrosine phosphorylation signaling cascades can switch protein functions to benefit cancer cells [73]. Caspase-8 phosphorylation has been investigated over the past twenty years as one of the mechanisms responsible for Caspase-8 enzymatic inactivation, promoting in contrast the acquisition of new protein functions critical for cancer development and sustainment [25,26,30]. The phosphorylation of Caspase-8 can occur on serine, threonine and tyrosine residues, resulting ultimately in the modulation of its canonical enzymatic activity. Several different kinases have been proposed over the years to affect Caspase-8 function (Table 1).

In this regard, it has been demonstrated that CDK1-dependent Caspase-8 phosphorylation on Serine 387 (Ser387) in the p10 catalytic subunit prevents procaspase-8 cleavage and maturation, resulting ultimately in the inhibition of apoptotic function [76,78,82]. In addition, RSK2 phosphorylation on Threonine 263 (Thr263) inhibits apoptotic function by acting on Caspase-8 stability [74,75]. Interestingly, p38-MAPK has also been proposed as a Caspase-8 modulator through its dependent phosphorylation on Serine 347 (Ser 347) [80] and Serine 364 (Ser 364) [8].

In recent years, studies have focused on the role of tyrosine phosphorylation of Caspase-8, mostly promoted by Src family non-receptor tyrosine kinases (SFKs). Compared to other caspases, Caspase-8 has a higher number of tyrosine residues (18), mostly located in the catalytic region [30]. Importantly, our group provided the identification of the first tyrosine residue on Caspase-8 undergoing phosphorylation: Tyr380 has been demonstrated to be phosphorylated by the non-receptor tyrosine kinase Src [25]. High endogenous phosphorylation levels on Tyr380 of Caspase-8 were observed in those contexts in which Src is aberrantly active, such as colon and hepatic tumors and glioblastoma, where Caspase-8 rewires its apoptotic and oncosuppressive function towards pro-tumoral functions [25,46]. 

Lyn, another Src family kinase, has been demonstrated to phosphorylate several tyrosine residues on Caspase-8 (Tyr397 and Tyr465), inhibiting its apoptotic function [79]. The hyperactivation of Lyn kinase plays an anti-apoptotic role in the regulation of neutrophil apoptosis during sepsis; together, with the knowledge that Lyn is frequently hyperactivated in myeloid and B cell malignancies [84] contributing to defective apoptosis, this reinforce the idea that tyrosine phosphorylation inhibits apoptotic cell death favoring tumoral cells in different cancers [25,46,49,84].

### 4.2. Src Kinase-Dependent Phosphorylation of Caspase-8 on Tyr380

The Src-dependent phosphorylation of Caspase-8 on Tyr380 inhibits its apoptotic function [25]. NMR spectroscopy demonstrated that Tyr380 phosphorylation significantly impinges on autoprocessing and the full activation of Caspase-8, reducing the rate of cleavage and thus explaining the inhibition of apoptotic function [85]. One likely reason behind this inhibition is that the large and negatively charged phosphate group may prevent the recognition of nearby cleavage sites (D374 and D384). In addition, as in living cells tyrosine phosphorylation create novel binding sites for cellular proteins, we can speculate that Tyr380 phosphorylation may lead to new protein–protein interactions and mask the cleavage sites [30].

The Src-dependent phosphorylation of Caspase-8 on Tyr380 has been shown to inhibit Caspase-8-dependent apoptosis in colon cancer cells and promote cell migration in neuroblastoma cell lines [25,31,48]. In line with this, Tyr380 phosphorylation promotes in vitro cell transformation in glioblastoma and hepatocellular carcinoma cellular models [49]. Importantly, it has been demonstrated that Caspase-8 is able to sustain in vitro transformation and resistance to anoikis independently of its enzymatic activity. In addition, increased Src activity observed in hypoxic conditions is strongly correlated with higher Caspase-8 phosphorylation, suggesting a functional link between the two proteins and giving tumor cells a selective advantage to sustain their growth in unfavorable conditions [49]. The mechanism through which Caspase-8 sustains tumorigenicity is still under investigation. Increasing evidence suggests that Caspase-8 may acquire a role as scaffold protein upon phosphorylation, as described by Keller et al. Actually, several studies have demonstrated the interaction of Caspase-8 with other proteins, mainly through the SH2 domain, in most cases helping and sustaining the activation and propagation of downstream signaling [48].

Studies from our and other laboratories demonstrated a physical interaction between Caspase-8 and Src, occurring specifically with Src homology 2 domain of the kinase, after Caspase-8 Tyr380 phosphorylation; this was responsible not only for the enzymatic inactivation of Caspase-8 but also for its different intracellular localization. Indeed, upon Src constitutive activation or EGF-mediated activation, Caspase-8 is associated with the cellular membrane, allowing sustained migration in cancer cells [47]. 

The Src-dependent phosphorylation on Tyr380 also allows Caspase-8 to form the FAK–Caspase-8–Calpain complex to promote cell migration and metastasis [86]. 

Importantly, previous findings demonstrated that Caspase-8 plays a critical role in promoting epidermal growth factor (EGF) signaling, resulting in ERK 1/2 activation [87]. This role is ensured by the ability of the DEDs domain to physically associate with Src. Intriguingly, it has been suggested that the interaction between the two proteins promotes the Src “open conformation”, which matches with its active state, thus supporting Caspase-8 role as a Src modulator [30].

It has been demonstrated that, upon EGF stimulation, Caspase-8 and active Src co-immunoprecipitate, also suggesting a role for Tyr380 in this context [87], and the existence of a crosstalk between receptor and non-receptor tyrosine kinase signaling and Caspase-8, especially in pathological contexts such as cancer; however, the precise mechanism deserves further elucidation.

We recently demonstrated that in glioblastoma cellular models, Caspase-8 phosphorylation, independent of its enzymatic activity, sustains NF-κB activation and translocation into the nucleus promoting the expression of its target inflammatory cytokines. Additionally in this context, Caspase-8 forms a multiprotein complex. Of note, Src-dependent Tyr380 phosphorylation promotes the interaction of Caspase-8 with Src and, more intriguingly, with the NFκB p65 protein, its upstream kinase IKKα/β and its inhibitor Iκbα [46]. 

Overall, these data reinforce the idea that Caspase-8 phosphorylation sustains tumor growth through the upregulation of the inflammatory pathway concurring with the establishment of a pro-angiogenic state and sustaining resistance to therapy (Figure 2) [6,46].

## 5. Role of RTK Signaling in the Modulation of Src-Dependent Phosphorylation of Tyr380 

Receptor tyrosine kinases are frequently aberrantly hyperactivated in cancer, resulting in the deregulation of intracellular signaling involving downstream non-receptor tyrosine kinases such as Src, and other Src-family kinases (SFKs), which act as central hubs to propagate deregulated and redundant upstream signals [88,89]. 

Despite the increasing amount of data supporting a role for tyrosine phosphorylation in modulating Caspase-8 activity, little is still known about a clear crosstalk between cancer-related RTKs deregulation, Caspase-8 phosphorylation and its protumor role. 

Deregulated RTKs in cancer result in the constitutive upregulation of intracellular signals among which are several kinases, including Src and Abl non-receptor tyrosine kinases [90]. Previous studies suggest that unlike Src, the Abl kinase fails to phosphorylate Caspase-8 on Tyr380 [25]. Importantly, EGF stimulation may enhance Tyr380 phosphorylation [25], supporting the hypothesis of a link between the hyperactivation of RTKs, the constitutive activation of Src and Caspase-8 phosphorylation on Tyr380. In line with this, we can therefore speculate that the rewiring of Caspase-8 function in cancer may be a direct consequence of the ability of constitutively active RTKs to promote the hyperactivation of Src family tyrosine kinases and therefore drive Caspase-8 phosphorylation on Tyr380 (Figure 2). Further studies are needed to deepen this issue and to clarify the link between RTKs and Caspase-8 non-apoptotic functions in cancers.

## 6. Conclusions

Caspase-8 expression varies widely among cancer subtypes [8,9]. Tumors such as medulloblastoma, neuroblastoma and small cell lung cancer, decrease Caspase-8 expression as a way of escaping the apoptotic form of death that regulates healthy tissue homeostasis [18,19,20,21]. Conversely, tumors such as glioblastoma, pancreatic cancer, head and neck cancer display unchanged or upregulated levels of Caspase-8 [12,41,91]. Tumors can retain or even upregulate Caspase-8 expression thanks to inactivating mutations or phosphorylation events that impinge on its enzymatic activity and apoptotic function. Indeed, in addition to CASP-8 inactivating mutations that can inhibit its proteolytic activity [13,16,92,93], Caspase-8 phosphorylation represents another important mechanism to preserve Caspase-8 expression in cancer. Importantly, not only does phosphorylation impinge on the catalytic activity of Caspase-8, thereby disrupting its apoptotic function [25], but intriguingly, these events can also promote cancer progression by enhancing cell motility, migration, inflammation, neoangiogenesis and resistance to genotoxic stress [10,28,46,67]. Most of the literature examining Caspase-8 non-canonical roles has so far focused mainly on the Src kinase-mediated phosphorylation on Tyr380 as the principal mechanism responsible for switching Caspase-8 fate from an apoptotic protein to a tumor helper. The molecular mechanisms through which this phosphorylation can affect tumor growth are still largely obscure. Future studies will clarify how Tyr380 phosphorylation may affect the interaction of Caspase-8 with other SH2-domain containing proteins, as previously suggested [48], and eventually promote the assembly of novel multiprotein complexes.

In addition, it will be interesting to investigate whether the expression and activity of tyrosine phosphatases may impinge on Tyr380 phosphorylation and contribute to modulating Caspase-8 activity and function in cancer.

The observation that RTKs are commonly deregulated in cancer, along with their well-known role as Src activators, suggests the hypothesis of crosstalk between RTKs aberrant signaling and the modulation of Caspase-8 function in cancer. Future experiments will clarify this issue. Despite the central role of RTKs and of Tyr380, many other sites have been found to be phosphorylated on Caspase-8 and their functions and implications have not yet been fully clarified. This review has summarized the current literature state of the art on the non-canonical roles of Caspase-8 and its phosphorylated sites are presented in Table 1. The kinases responsible for carrying out Caspase-8 modifications are kinases typically known to be deregulated in cancer and many have already been extensively studied and used as targets of FDA-approved inhibitors. A better understanding of the significance of their phosphorylation of Caspase-8 will help repurpose existing inhibitors to alter the balance between Caspase-8 canonical and non-canonical roles and switch it back to its beneficial and antitumor apoptotic function.

## Figures and Tables

**Figure 1 cancers-15-03271-f001:**
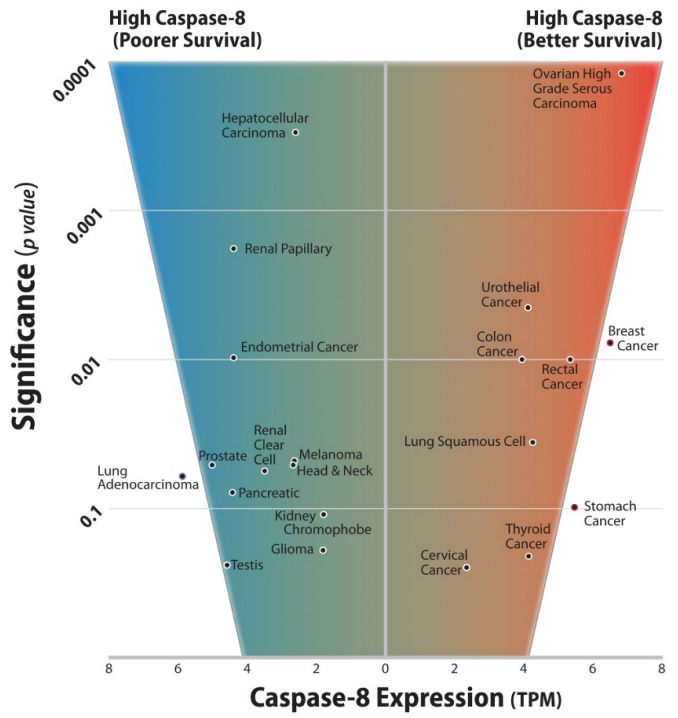
Twenty cancer databases from The Cancer Genome Atlas (TCGA) with overall survival data were queried to determine if the impact of Caspase-8 expression was a significant predictor of improved or worsened overall survival by Kaplan–Meier analysis. Using optimal cutoffs for Caspase-8 expression to divide the populations, each of the twenty cancers was found to trend towards significance or to demonstrate significance with respect to overall patient survival. The significance is shown plotted against the mean Caspase-8 transcript expression for the tumor type (tpm). Tumors were split roughly into two groups. The first group (right side of the panel) showed increased survival associated with increased Caspase-8 transcripts. The second group (left side of the panel) showed decreased survival associated with Caspase-8. No groups yielded a p value of greater than 0.3, raising the intriguing possibility that Caspase-8 is either pro- or anti-tumor, but rarely “neutral”. The results are generally consistent with the differing roles played by Caspase-8 in tumor promotion vs. tumor cell death.

**Figure 2 cancers-15-03271-f002:**
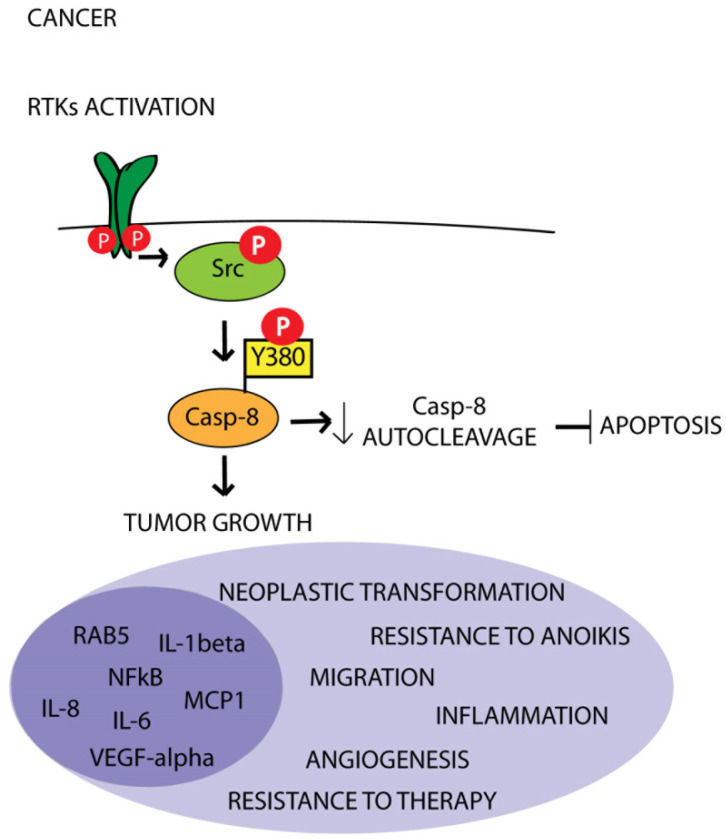
Schematic illustration of the interplay between RTKs, Src kinase and Caspase-8 in cancer. Aberrant activation of RTKs sustains Src activity, which in turn promotes Caspase-8 phosphorylation on Tyr380 (Y380), thereby inhibiting Caspase-8 apoptotic function and promoting its protumor functions.

**Table 1 cancers-15-03271-t001:** List of kinases able to phosphorylate Caspase-8 and result in the modulation of its canonical function. For residue numbering, Caspase-8 alpha-1 isoform was used as reference; when other isoforms are considered in the reference papers the respective alpha-1 numbering is reported in parentheses. For every kinase inhibitor currently in use, clinical trials are indicated.

Kinase	FDA Approved Kinase Inhibitors	Residue	Effect on Caspase-8	Reference	Clinical Trial
** RSK2 **	PMD-026	Thr263	Induces Caspase-8 ubiquitination Inhibits Fas-induced apoptosis	[8,74,75]	Metastatic Breast Cancer (NCT04115306 2022)
** PLK3 **	BI6727 (Volasertib)	Thr273	Promotes DISC-induced activation of Caspase-8	[6,76,77]	Acute Myeloid Leukemia (NCT01721876 2023, NCT00804856 2021) Solid Tumors (NCT02273388 2021) Ovarian Cancer (NCT01121406 2015)
** PLK1 **	BI2536	Ser305	Blocks Fas-induced apoptosis	[6,8,78]	NSCLC (NCT00376623 2022) Pancreatic Cancer (NCT00710710 2022) SCLC (NCT00412880 2022)
** SRC **	Dasatinib, Saracatinib, TPX0046	Tyr310 (Tyr 293)	Allows interaction with SHP-1, facilitating apoptosis	[79]	Breast Cancer (NCT01216176 2019) Leukemia (NCT00306202 2021) NSCLC (NCT00459342 2021) Metastatic Breast Cancer (NCT01306942 2023) Solid Tumors (NCT01445509 2023, NCT04161391 2023) Prostate Cancer (NCT00513071 2018)
** p38-MAPK **	LYN2228820, LYN3007113	Ser347	Inhibits apoptosis	[80]	Advanced Cancer (NCT01393990 2020, NCT01463631 2018) Glioblastoma (NCT02364206 2019) Ovarian Cancer (NCT01663857 2019)
** p38 **	LYN2228820, LYN3007113	Ser364 (Ser347)	-	[8]	Advanced Cancer (NCT01393990 2020, NCT01463631 2018) Glioblastoma (NCT02364206 2019) Ovarian Cancer (NCT01663857 2019)
** SRC/LYN **	Dasatinib, Saracatinib, TPX0046	Tyr380	Impairs apoptosis and increases cell motility, inflammation and tumorigenesis	[6,25,46,47,49,81]	Breast Cancer (NCT01216176 2019) Leukemia (NCT00306202 2021) NSCLC (NCT00459342 2021) Metastatic Breast Cancer (NCT01306942 2023) Solid tumors (NCT01445509 2023, NCT04161391 2023) Prostate Cancer (NCT00513071 2018)
** CDK1, ERK1/2 **	PD0332991, P276-00	Ser387	Allows subsequent phosphorylation on Ser305 Reduces D384 cleavage in p10	[8,76,82]	Solid Cancer (NCT01037790 2021, NCT00407498 2009)
** LYN **	Bafetinib, Dasatinib, Rituximab	Tyr397 (Tyr380)	Resistant to cleavage Inhibits apoptosis	[79]	Glioma (NCT01234740 2018) Leukemia (NCT00438854 2017) Lymphoma (NCT00788684 2022, NCT01775631 2017)
** LYN **	Bafetinib, Dasatinib, Rituximab	Tyr448	-	[6,79]	Glioma (NCT01234740 2018) Leukemia (NCT00438854 2017) Lymphoma (NCT00788684 2022, NCT01775631 2017)
** LYN **	Bafetinib, Dasatinib, Rituximab	Tyr465 (Tyr450)	Resistant to cleavage Inhibits apoptosis	[79,83]	Glioma (NCT01234740 2018) Leukemia (NCT00438854 2017) Lymphoma (NCT00788684 2022, NCT01775631 2017)

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
