# Peer review of "Caspase-8 and Tyrosine Kinases: A Dangerous Liaison in Cancer"

_cancers, 2023, doi:10.3390/cancers15133271_

Round 1

Reviewer 1 Report

The review article is well organized and aims to conclude current research in the field of oncology, specifically focusing on the Caspase-8 canonical and non-canonical roles. However, certain aspects must be highlighted and revised to present the manuscript's impact and findings effectively. It is recommended to include additional literature that explains the rationale and objectives of the study to provide a clear understanding for the readers. The authors should explain further and elaborate on the rationale behind their hypotheses.

Major comments:

  1. In accordance with section 4.1 of the manuscript, it should be elaborated on whether phosphorylation is the sole affected epigenetic mechanism (line 308).
  2. Additionally, the authors should provide more information about the concept of dephosphorylation of Caspase-8. Moreover, it would be valuable to determine whether dephosphorylation also leads to the dual roles of caspase 8.
  3. Including more details on the cell cycle arrest and inflammatory aspects of caspase-8 phosphorylation would establish a foundation for understanding the functional relevance shift.
  4. Figure 2 should be enhanced to provide a more conclusive summary of the interplay between RTKs, SRC kinase, and Caspase-8, thereby facilitating a better comprehension of the signaling pathways and molecules affected by these changes.
  5. In lines 234-243, the authors imply the effects of p53. It would be helpful to explain further the difference in changes between the p53 wild type and mutant in order to understand the concept of cell cycle arrest.
  6. By mentioning the role of RTKs signaling in the modulation of Src-dependent phosphorylation of Tyr380 (lines 406-423), the authors may be implying an association between the change in function of Tyr380 in accordance with tumor type and the modulation of Src-dependent phosphorylation of Tyr380 as discussed in lines 336-339.
  7. Which other signaling effectors influence the roles of Caspase-8 in both canonical and non-canonical capacities?
  8. In conclusion, how close are caspase inhibitors to being used clinically, considering their rarely neutral behavior as described by the authors in line 128?
  9. In line 346, the term "different cancers" should be cross-referenced with previously published literature.
  10. To better understand the various roles played by caspase-8 under different conditions, it is advisable to summarize the information using a figure illustrating the changes in cell cycle regulators and their influence on caspase-8-dependent functions.
  11. Since caspase-8 is retained in tumors, suggesting the presence of alternative mechanisms that cancer cells may exploit for their benefit, the authors should express their views on specific caspase-8 mutations in the context of gain or loss of function.
  12. The conclusion and future directions sections should be more insightful and comprehensively described.

Minor comments:

  1. More literature should be cited to support and interpret the provided information.
  2. It is recommended to proofread the manuscript for grammatical errors and formatting.
  3. The authors should use online portals such as iThenticate to cross-check the manuscript for phrase duplications.

  1. It is recommended to proofread the manuscript for grammatical errors and formatting.
  2. The authors should use online portals such as iThenticate to cross-check the manuscript for phrase duplications.

Reviewer 2 Report

The article “Caspase-8 and tyrosine kinases: a dangerous liaison in cancer” by Contadini et al, presents a comprehensive review on Caspase-8 and its phosphorylation in cancer progression. The manuscript is written well and covers relevant literature on this topic. One suggestion I have for authors is to include an illustration about canonical Caspase- 8 signaling cascade in normal cells and its dysregulated signaling cascade in cancel cells. Does phosphorylation cause any conformational changes in Caspase-8? Wonder how exactly it affects its enzymatic activity?

Good. Some proof reading is suggested
